# A Refined Margin Distribution Analysis for Forest Representation Learning

**Shen-Huan Lyu, Liang Yang, Zhi-Hua Zhou**
National Key Laboratory for Novel Software Technology
Nanjing University, Nanjing, 210023, China
{lvsh,yangl,zhouzh}@lamda.nju.edu.cn

## Abstract

In this paper, we formulate the forest representation learning approach named casForest as an additive model, and show that the generalization error can be bounded by $\mathcal{O}(\ln m/m)$, when the margin ratio related to the margin standard deviation against the margin mean is sufficiently small. This inspires us to optimize the ratio. To this end, we design a margin distribution reweighting approach for the deep forest model to attain a small margin ratio. Experiments confirm the relation between the margin distribution and generalization performance. We remark that this study offers a novel understanding of casForest from the perspective of the margin theory and further guides the layer-by-layer forest representation learning.

## 1   Introduction

In recent years, deep neural networks have achieved excellent performance in many application scenarios such as face recognition and automatic speech recognition [19]. It is well known that deep neural networks are difficult to be interpreted. This severely restricts the development of deep learning in application scenarios where the interpretability of the model is crucial. Moreover, deep neural networks are data-hungry and the performance will degrades significantly when the size of the training data is not big enough [12, 20]. In real-world tasks, due to the high cost of data collection and labeling, the amount of labeled training data may be insufficient for deep neural networks.

In such a situation, conventional learning methods such as support-vector machines (SVMs) [7], random forests (RFs) [3], gradient boosting decision trees (GBDTs) [15, 5], etc., are still good choices. By realizing that the essence of deep learning lies in the *layer-by-layer processing*, *in-model feature transformation*, and *sufficient model complexity*, recently Zhou & Feng [32, 33] propose the deep forest model and the gcForest algorithm that incorporate *forest representation learning*. It can achieve excellent performance on a broad range of tasks, and even perform well on small or middle-scale data. Later on, a more efficient improvement is made by Pang et al. [21]. Feng & Zhou [13] show that forests are able to do auto-encoder which was considered as a specialty of neural networks. The tree-based multi-layer model can do hierarchical distributed representation learning which was thought to be a special feature of neural networks [14]. Utkin & Ryabinin [25] propose a Siamese deep forest as an alternative to the Siamese neural network for metric learning tasks.

The cascade forest (abbr. casForest) structure plays an important role in Deep Forest, and it is crucial for the layer-by-layer processing. This paper attempts to explain the benefits of casForest from the perspective of the *margin theory*.

## 1.1 Our Results

In Section 2, we formulate casForest (see the structure in Figure 1) as an additive model (the additive casForest model) to optimize the margin distribution:

$$F(x) = \sum_{t=1}^{T} \alpha_t h_t(x), \tag{1}$$

where $\alpha_t$ is a scalar determined by the margin distribution loss function $\ell_{md}$. The input of each random forests block function $\phi_t$ is the *raw feature* $x$ and the $(t-1)$-th *augmented feature* $f_{t-1} = \sum_{l=1}^{t-1} \alpha_l h_l$:

$$h_t(x) = \phi_t\left([x, f_{t-1}(x)]\right) = \phi_t\left(\left[x, \sum_{l=1}^{t-1} \alpha_l h_l(x)\right]\right), \tag{2}$$

so that the $t$-layer casForest model $h_t \in \mathcal{H}_t$ is defined by such a recursive formula. Unlike all the weak classifiers of traditional boosting are chosen from the same hypothesis set $\mathcal{H}$, the hypothesis set of the $t$-layer casForest model contains the $(t-1)$-layer[1], i.e., $\mathcal{H}_{t-1} \subset \mathcal{H}_t, \forall t \geq 2$.

In Section 3, we provide a margin distribution upper bound for the generalization error of the additive model above:

$$\Pr_{\mathcal{D}}[yF(x) < 0] - \Pr_{S}[yF(x) < r] \leq \frac{\ln\sum_{t=1}^{T} \alpha_t \left|\mathcal{H}_t\right|}{r^2} \cdot \frac{\ln m}{m} + \lambda\sqrt{\frac{\ln\sum_{t=1}^{T} \alpha_t \left|\mathcal{H}_t\right|}{r^2} \cdot \frac{\ln m}{m}}, \tag{3}$$

where $m$ is the size of training set, $r$ is a margin parameter, $\lambda = \sqrt{\frac{\mathrm{Var}[yF(x)]}{\mathbb{E}_S^2[yF(x)]}}$ is a ratio related to the margin standard deviation against the margin mean, and $yF(x)$ denotes the margin of the sample $x$.

Inspired by our theoretical result, we propose an effective algorithm named margin distribution Deep Forest (see mdDF in Algorithm 2) to encourage optimizing the margin ratio. Extensive experiments validate that mdDF can effectively improve the performance on classification tasks, especially for categorical and mixed modeling tasks.

## 1.2 Related Work

**Deep Forest.** Deep Forest [32, 33] is a non-neural network deep learning model which builds upon decision trees and does not rely on BP algorithm and gradient-based approach. The earliest deep forest algorithm gcForest [33], is constructed by the multi-grained scanning operation and the casForest structure. The multi-grained scanning operation aims to deal with the raw data with spatial or sequential relations. The casForest structure aims at the layer-by-layer processing with in-model feature transformation. It can be viewed as an ensemble approach that utilizes almost all categories of well-known strategies for diversity enhancement, e.g., input feature manipulation and output representation manipulation [30].

**Margin theory.** The margin theory was used by Schapire et al. [24] to explain the resistance of AdaBoost to overfitting, but then attacked almost to death by the construction of the Arc algorithm by Breiman [2]. Later on, it was found that the empirical attack to margin theory of Adaboost might be misleading [22], and many theoretical studies tried to get more understanding, ended by Gao & Zhou [16]. They finally proved that the margin distribution, which can be improved by increasing the margin mean while decreasing the margin variance, is crucial to the performance of AdaBoost. This has inspired the birth of a series of new statistical learning algorithms named ODM [31, 28, 29].

## 2 Cascade Forest

In Figure 1, the casForest structure is composed of stacked entities named random forests blocks $\phi_t$s. Each random forests block consists of several forest modules, e.g., commonly random forests

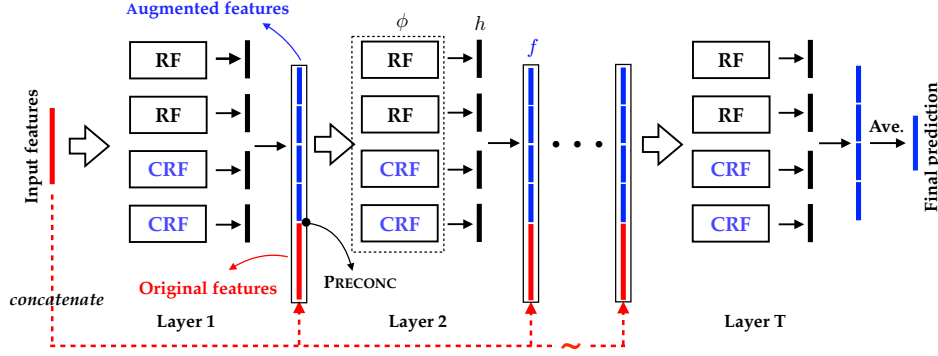

Figure 1: The standard cascade structure of the deep forest model [33] can be viewed as a layer-by-layer process. This feature augmentation can achieve feature enrichment by concatenating the prediction vector with the input feature vector, which is named "PRECONC".

(abbr. RF) [3] and completely-random forests (abbr. CRF) [32]. Suppose $f_1$ denotes the function of the first-layer forests, then given the input $x$ to the first layer, the input to the second layer will be $[x, f_1(x)]$, where $[a, b]$ denotes the concatenation of $a$ and $b$ to form a feature vector. Considering that the $f_1(x)$ is the prediction from the first layer, we name this process as "PRECONC" (PREdiction CONCatenation), which is crucial for the feature learning process in deep forest. PRECONC is different from the stacking operation [27, 1] in traditional ensemble learning, where the second-level learners act on the prediction space composed of different base learners, whereas the information of the original input feature space is ignored. Using the stacking operation with more than two layers would seriously suffer from overfitting, and cannot enable a deep model. In this paper we do not study the factors which enable deep forest to become a deep model, only focus on the cascade structure.

Firstly we formulate casForest as an additive model in this section. We consider training and test samples generated i.i.d. from distribution $\mathcal{D}$ over $\mathcal{X} \times \mathcal{Y}$, where $\mathcal{X} \in \mathbb{R}^n$ is the input space and $\mathcal{Y} \in \{1, 2, \ldots, s\}$ is the output space. We denote a training set of $m$ samples drawn from $\mathcal{D}^m$ by $S$.

The casForest model can be formalized as follows. We use a quadruple form $(\boldsymbol{\phi}, \boldsymbol{f}, \boldsymbol{\mathcal{D}}, \boldsymbol{h})$ where

- **Forest block:** $\boldsymbol{\phi} = (\phi_1, \phi_2, \ldots, \phi_T)$, where $\phi_t$ denotes the function computed by the random forests block in the $t$-th layer which is defined by (4);
- **casForest:** $\boldsymbol{h} = (h_1, h_2, \ldots, h_T)$, where $h_t$ denotes the $t$-layer casForest model defined by (5), and $h_t$ drawn from the hypothesis set $\mathcal{H}_t$;
- **Augmented feature:** $\boldsymbol{f} = (f_1, f_2, \ldots, f_T)$, where $f_t$ denotes the output in the $t$-th layer, which is defined by (6);
- **Sample distribution:** $\boldsymbol{\mathcal{D}} = (\mathcal{D}_1, \mathcal{D}_2, \ldots, \mathcal{D}_T)$, where $\mathcal{D}_t$ is the updated sample distribution in the $t$-th layer, and $\mathcal{D}_1 = \mathcal{D}$.

$\phi_t$ is the function returned by the random forests block (Algorithm 1). The input of the algorithm is the *raw training sample* $S = \{(x_1, y_1), \ldots, (x_m, y_m)\}$, the *augmented feature* from the previous layer $f_{t-1}(x_i), i \in [m]$, and the reweighting distribution $\mathcal{D}_t$:

$$\phi_t = \begin{cases} \mathcal{A}_{\text{rfb}}\left([x_i; y_i]_{i=1}^m, \mathcal{D}_1\right) & t = 1, \\ \mathcal{A}_{\text{rfb}}\left([x_i, f_{t-1}(x_i); y_i]_{i=1}^m, \mathcal{D}_t\right) & t > 1. \end{cases} \tag{4}$$

Using these random forests block functions $\phi_t$s, we can define the $t$-layer casForest model as:

$$h_t(x) = \begin{cases} \phi_t(x) & t = 1, \\ \phi_t\left([x, f_{t-1}(x)]\right) & t > 1, \end{cases} \tag{5}$$

$f_t : \mathcal{X} \to \mathcal{C}$ is defined as follows:

$$f_t(x) = \begin{cases} \alpha_t h_t(x) & t = 1, \\ \alpha_t h_t(x) + f_{t-1}(x) & t > 1, \end{cases} \tag{6}$$

---
**Algorithm 1** Random forests block $\mathcal{A}_{\text{rfb}}$ [33]
---
**Input:** A training set $S$ drawn from $\mathcal{D}_t$ and the augmented feature $f_{t-1}(x_i), \forall i \in [m]$.
**Output:** The function computed by the random forests block in the $t$-th layer: $\phi_t$.
1: Divide $S$ to $k$-fold subsets $\{S_1, \ldots, S_k\}$ randomly.
2: **for** $S_i$ in $\{S_1, S_2, \ldots, S_k\}$ **do**
3:    Using $S/S_i$ to train two random forests and two completely random forests.
4:    Compute the prediction rate $p_t^i(j)$ for the $j$-th leaf node generated by $S/S_i$.
5:    $\phi_t([x, f_{t-1}(x)]) \leftarrow \mathbb{E}_j[p_t^i(j)]$, for any training sample $(x, y) \in S_i$.
6: **end for**
7: $\phi_t([x, f_{t-1}(x)]) \leftarrow \mathbb{E}_{i,j}[p_t^i(j)]$, for any test sample $(x, y) \in \mathcal{D}$.
8: **return** The function computed by the random forests block in the $t$-th layer: $\phi_t$.
---

where $\alpha_t$ and $\mathcal{D}_t$ need to be optimized and updated.

Here, we find that the $t$-layer casForest model is defined by a recursive formula:

$$h_t(x) = \phi_t([x, f_{t-1}(x)]) = \phi_t\left(\left[x, \sum_{l=1}^{t-1} \alpha_l h_l(x)\right]\right). \tag{7}$$

Unlike all the weak classifiers of AdaBoost which are chosen from the same hypothesis set $\mathcal{H}$, the hypothesis set of the $t$-layer casForest model contains that of the $(t-1)$-layer, similar to the hypothesis sets of the deep neural networks (DNNs) at different depths, i.e., $\mathcal{H}_{t-1} \subset \mathcal{H}_t, \forall t \geq 2$.

The PRECONC process is difficult to analyze. For simplicity, here we do not consider the influence of the feature augmentation process though it is very crucial for deep forest. Instead, we only consider the hypotheses based on the original feature space, and thus the entire additive cascade model $\tilde{F} : \mathcal{X} \to \mathcal{Y}$ is defined as follows:

$$\tilde{F}(x) = \tilde{\sigma}(F(x)) = \underset{j \in \{1,2,\ldots,s\}}{\arg\max} \left[\sum_{t=1}^{T} \alpha_t h_t^j(x)\right], \tag{8}$$

where $F(x)$ is the final prediction vector of the casForest model for classification and $\tilde{\sigma}$ denotes a map from average prediction score vector to a label.

With such a simplicity, the casForest structure has relation to Cortes et al. [8, 9] and Huang et al. [17]. However, in the next section we will see that we prove that the generalization error of casForest can be bounded by $\mathcal{O}(\ln m/m + \lambda\sqrt{\ln m/m})$, when the margin ratio related to the margin standard deviation against the margin mean is sufficient small. This bound is tighter than the generalization bound $\mathcal{O}(\ln m/m)$ for Deep Boosting [8, 9, 17].

## 3   Generalization Analysis

In this section, we analyze the generalization error to understand the sample complexity of the casForest model. For simplicity, we consider the binary classification[2] task. We define the strong classifier (the $T$-layer casForest model) as $F(x) = \sum_{t=1}^{T} \alpha_t h_t(x)$, i.e., casForest is formulated as an additive model. Now we define the margin for sample $(x, y)$ as $yF(x) \in [-1, 1]$, which implies the confidence of prediction. We assume that the hypothesis set $\mathcal{H}$ of base classifiers $\{h_1, h_2, \ldots, h_T\}$ can be decomposed as the union of $T$ families $\mathcal{H}_1, \mathcal{H}_2, \ldots, \mathcal{H}_T$ ordered by increasing complexity, where $\forall t \geq 1, \mathcal{H}_t \subset \mathcal{H}_{t+1}$ and $h_t \in \mathcal{H}_t$. Remarkably, the complexity term of our bound admits an explicit dependency in terms of the mixture coefficients defining the ensembles. Thus, the ensemble family we consider is $\mathcal{F} = \text{conv}(\bigcup_{t=1}^{T} \mathcal{H}_t)$, which is the family of functions $F(x)$ of the form $F(x) = \sum_{t=1}^{T} \alpha_t h_t(x)$, where $\boldsymbol{\alpha} = (\alpha_1, \ldots, \alpha_T)$ is in the simplex $\Delta$.

For a fixed $\boldsymbol{g} = (g_1, \ldots, g_T)$, any $\boldsymbol{\alpha} \in \Delta$ defines a distribution over $\{g_1, \ldots, g_T\}$. Sampling from $\{g_1, \ldots, g_T\}$ according to $\boldsymbol{\alpha}$ and averaging leads to functions $G = \frac{1}{n}\sum_{i=1}^{T} n_t g_t$ for some

$\mathbf{n} = (n_1, \ldots, n_T)$, with $\sum_{t=1}^T n_t = n$, and $g_t \in \mathcal{H}_t$. For any $\mathbf{N} = (N_1, \ldots, N_T)$ with $|\mathbf{N}| = n$, we consider the family of functions

$$\mathcal{G}_{\mathcal{F},\mathbf{N}} = \left\{ \frac{1}{n} \sum_{k=1}^T \sum_{j=1}^{N_k} g_{k,j} \,\middle|\, \forall (k,j) \in [T] \times [N_k], g_{k,j} \in \mathcal{H}_k \right\}, \tag{9}$$

and the union of all such families $\mathcal{G}_{\mathcal{F},n} = \bigcup_{|\mathbf{N}=n|} \mathcal{G}_{\mathcal{F},\mathbf{N}}$. For a fixed $\mathbf{N}$, the size of $\mathcal{G}_{\mathcal{F},\mathbf{N}}$ can be bounded as follows: $\ln |\mathcal{G}_{\mathcal{F},\mathbf{N}}| \leq n \ln \sum_{t=1}^T \alpha_t |\mathcal{H}_t|$. Our margin distribution theory is based on a bound based on the margin mean and a Bernstein-type bound follows:

**Lemma 1.** ([16]) *For $F = \sum_{t=1}^T \alpha_t h_t \in \mathcal{F}$ and $G \in \mathcal{G}_{\mathcal{F},n}$, we have*

$$\Pr_{S,\mathcal{G}_{\mathcal{F},n}} [yG(x) - yF(x) \geq \epsilon] \leq \exp \left( \frac{-n\epsilon^2}{2 - 2\mathbb{E}_S^2[yF(x)] + 4\epsilon/3} \right). \tag{10}$$

**Lemma 2.** ([16]) *For independent random variables $X_1, X_2, \ldots, X_m (m \geq 5)$ with values in $[0,1]$, and for $\delta \in (0,1)$, with probability at least $1 - \delta$ we have*

$$\frac{1}{m} \sum_{i=1}^m \mathbb{E}[X_i] - \frac{1}{m} \sum_{i=1}^m X_i \leq \sqrt{\frac{2\hat{V}_m \ln(2/\delta)}{m}} + \frac{7 \ln(2/\delta)}{3m} \tag{11}$$

*where $\hat{V}_m = \sum_{i \neq j} (X_i - X_j)^2 / 2m(m-1)$*

Since the gap between the margin of strong classifier $yF(x)$ and that in the union family $\mathcal{G}_{\mathcal{F},\mathbf{N}}$ is bounded by a function related to the margin mean $\mathbb{E}_S[yF(x)]$, we can further obtain a margin distribution theorem as follows:

**Theorem 1.** *Let $\mathcal{D}$ be a distribution over $\mathcal{X} \times \mathcal{Y}$ and $S$ be a training set of $m$ samples drawn from $\mathcal{D}$. With probability at least $1 - \delta$, for $r > 0$, the strong classifier $F(x)$ (the $T$-layer casForest model) satisfies that*

$$\Pr_{\mathcal{D}}[yF(x) < 0] \leq \inf_{r \in (0,1]} \left[ \Pr_S[yF(x) < r] + \frac{1}{m^d} + \frac{3\sqrt{\mu}}{m^{3/2}} + \frac{7\mu}{3m} + \lambda \sqrt{\frac{3\mu}{m}} \right]$$

*where*

$$d = \frac{2}{1 - \mathbb{E}_S^2[yF(x)] + r/9} > 2, \ \mu = \ln m \ln(2 \sum_{t=1}^T \alpha_t |\mathcal{H}_t|)/r^2 + \ln \frac{2}{\delta}, \lambda = \sqrt{\frac{\mathrm{Var}[yF(x)]}{\mathbb{E}_S^2[yF(x)]}}.$$

*Proof.* For $F = \sum_{t=1}^T \alpha_t h_t \in \mathcal{F}$ and $G \in \mathcal{G}_{\mathcal{F},n}$, we have $\mathbb{E}_{G \in \mathcal{G}_{\mathcal{F},n}}[G] = F$. For $\beta > 0$, the Chernoff's bound gives

$$\Pr_{\mathcal{D}} [yF(x) < 0] = \Pr_{D,\mathcal{G}_{\mathcal{F},n}} [yF(x) < 0, yG(x) \geq \beta] + \Pr_{D,\mathcal{G}_{\mathcal{F},n}} [yF(x) < 0, yG(x) < \beta]$$

$$\leq \exp(-n\beta^2/2) + \Pr_{D,\mathcal{G}_{\mathcal{F},n}} [yG(x) < \beta]. \tag{12}$$

Recall that $|\mathcal{G}_{\mathcal{F},N}| \leq \prod_{t=1}^T |\mathcal{H}_t|^{N_t}$ for a fixed $N$. Therefore, for any $\delta_n > 0$, combining the union bound with the Lemma 2 guarantees that with probability at least $1 - \delta_n$ over sample $S$, for any $G \in \mathcal{G}_{\mathcal{F},N}$ and $\beta > 0$

$$\Pr_{D}[yG(x) < \beta] \leq \Pr_S[yG(x) < \beta] + \sqrt{\frac{2}{m} \hat{V}_m \ln \left( \frac{2}{\delta} \prod_{t=1}^T |\mathcal{H}_t|^{N_t} \right)} + \frac{7}{3m} \ln \left( \frac{2}{\delta} \prod_{t=1}^T |\mathcal{H}_t|^{N_t} \right) \tag{13}$$

$$\leq \Pr_S[yG(x) < \beta] + \sqrt{\frac{2n}{m} \hat{V}_m \sum_{i=1}^T \alpha_t \ln \left( \frac{2|\mathcal{H}_t|}{\delta} \right)} + \frac{7n}{3m} \sum_{i=1}^T \alpha_t \ln \left( \frac{2|\mathcal{H}_t|}{\delta} \right) \tag{14}$$

$$\leq \Pr_{S}[yG(x) < \beta] + \sqrt{\frac{2n}{m}\hat{V}_m \ln\left(\frac{2\sum_{i=1}^{T}\alpha_t|\mathcal{H}_t|}{\delta}\right)} + \frac{7n}{3m}\ln\left(\frac{2\sum_{i=1}^{T}\alpha_t|\mathcal{H}_t|}{\delta}\right) \tag{15}$$

where

$$\hat{V}_m = \sum_{i \neq j} \frac{(\mathbb{I}[y_iG(x_i) < \beta] - \mathbb{I}[y_jG(x_j) < \beta])^2}{2m(m-1)}, \tag{16}$$

The inequality (14) is a large probability bound when $n$ is large enough and inequality (15) is according to the Jensen's Inequality. Since there are $T$ at most $T^n$ possible $T$-tuples $N$ with $|N| = n$, by the union bound, for any $\delta > 0$, with probability at least $1 - \delta$, for all $G \in \mathcal{G}_{\mathcal{F},n}$ and $\beta > 0$:

$$\Pr_{D}[yG(x) < \beta] \leq \Pr_{S}[yG(x) < \beta] + \sqrt{\frac{2n}{m}\hat{V}_m \ln\left(\frac{2\sum_{i=1}^{T}\alpha_t|\mathcal{H}_t|}{\delta/T^n}\right)} + \frac{7n}{3m}\ln\left(\frac{2\sum_{i=1}^{T}\alpha_t|\mathcal{H}_t|}{\delta/T^n}\right) \tag{17}$$

Meantime, we can rewrite $\hat{V}_m$

$$\hat{V}_m = \sum_{i \neq j} \frac{(\mathbb{I}[y_iG(x_i) < \beta] - \mathbb{I}[y_jG(x_j) < \beta])^2}{2m(m-1)} \tag{18}$$

$$= \frac{2m^2 \Pr_S[yG(x) < \beta] \Pr_S[yG(x) \geq \beta]}{2m(m-1)} \tag{19}$$

$$= \frac{m}{m-1}\hat{V}_m^* \tag{20}$$

For any $\theta_1, \theta_2 > 0$, we utilize Chernoff's bound to get:

$$\hat{V}_m^* = \Pr_S[yG(x) < \beta] \Pr_S[yG(x) \geq \beta] \tag{21}$$

$$\leq 3\exp(-n\theta_1^2/2) + \Pr_S[yF(x) < \beta + \theta_1] \Pr_S[yF(x) \geq \beta - \theta_1] \tag{22}$$

$$\leq 3\exp(-n\theta_1^2/2) + \Pr_S[yF(x) < \beta + \theta_1 \,|\, \mathbb{E}_S[yF(x)] \geq \beta + \theta_1 + \theta_2] \tag{23}$$

$$\quad \cdot \Pr_S[yF(x) \geq \beta - \theta_1 | \mathbb{E}_S[yF(x)] \geq \beta + \theta_1 + \theta_2]$$

$$\leq 3\exp(-n\theta_1^2/2) + \frac{\text{Var}[yF(x)]}{\theta_2^2} \qquad \text{According to Chebyshev's Inequality}$$

$$\leq 3\exp(-n\theta_1^2/2) + \frac{\text{Var}[yF(x)]}{(\mathbb{E}_S[yF(x)] - \beta + \theta_1)^2} \simeq 3\exp(-n\theta_1^2/2) + \frac{\text{Var}[yF(x)]}{\mathbb{E}_S^2[yF(x)]} \tag{24}$$

where $\text{Var}[yF(x)] = \mathbb{E}_S[(yF(x))^2] - \mathbb{E}_S^2[yF(x)]$ is the variance of the margins.

From Lemma 1, we obtain that

$$\Pr_S[yG(x) < \beta] \leq \Pr_S[yF(x) < \beta + \theta_1] + \exp\left(\frac{-n\theta_1^2}{2 - 2\mathbb{E}_S^2[yF(x)] + 4\theta_1/3}\right) \tag{25}$$

Let $\theta_1 = r/6$, $\beta = 5r/6$ and $n = \ln m/r^2$, we combine (12)(15)(24)(25), the proof is completed. $\square$

**Remark 1.** From Theorem 1, we know that the gap between the generalization error and the empirical margin loss is generally bounded by the term $\mathcal{O}(\lambda\sqrt{\ln m/m} + \ln m/m)$, which is controlled by the ratio related to the margin standard deviation against the margin mean $\lambda$. This ratio implies that the larger margin mean and the smaller margin variance can reduce the generalization error of models properly, which is crucial to alleviating the overfitting problem. When the margin distribution is good enough (the margin mean is large and the margin variance is small), $\mathcal{O}(\ln m/m)$ will dominate the sample complexity. Then, this bound is tighter than the $\mathcal{O}(\sqrt{\ln m/m})$ rate as demonstrated in previous theoretical works about Deep Boosting [8, 9, 17].

**Algorithm 2** mdDF (margin distribution Deep Forest)
___
**Input:** Training set $S = \{(x_1, y_1), \ldots, (x_m, y_m)\}$ and random forests block algorithm $\mathcal{A}_{\mathrm{rfb}}$.
**Output:** The final additive cascade model $\tilde{F}$.
 1: Initialize $\alpha_0 \leftarrow 1, f_0 \leftarrow \emptyset$
 2: Initialize sample weights: $\mathcal{D}_1(i) \leftarrow \frac{1}{m}, \forall i \in [m]$
 3: **for** $t = 1, 2, \ldots, T$ **do**
 4:    $\phi_t \leftarrow$ the random forests block returned by $\mathcal{A}_{\mathrm{rfb}}([x_i, f_{t-1}(x_i); y_i]_{i=1}^m, \mathcal{D}_t)$.
 5:    $h_t(x_i) \leftarrow \phi_t([x_i, f_{t-1}(x_i)]), \forall i \in [m]$.
 6:    $\gamma_t(x_i) \leftarrow h_t^y(x_i) - \max_{j \neq y} h_t^j(x_i), \forall i \in [m]$.
 7:    $\alpha_t \leftarrow \underset{\alpha_t}{\arg\min} \, \mathbb{E}_S[\ell_{\mathrm{md}}(\sum_{l=1}^t \alpha_l \gamma_l(x))]$
 8:    $f_t(x_i) \leftarrow \alpha_t h_t(x_i) + f_{t-1}(x_i), \forall i \in [m]$.
 9:    $\mathcal{D}_{t+1}(i) \leftarrow \frac{\ell_{\mathrm{md}}\left(\sum_{l=1}^t \alpha_l \gamma_l(x_i)\right)}{\sum_{i=1}^m \ell_{\mathrm{md}}\left(\sum_{l=1}^t \alpha_l \gamma_l(x_i)\right)}, \forall i \in [m]$.
10: **end for**
11: **return** $\tilde{F} \leftarrow \underset{j \in \{1,2,\ldots,s\}}{\arg\max} \left[\sum_{t=1}^T \alpha_t h_t^j\right]$.
___

**Remark 2.** As for the overfitting risk of the model (due to the large complexity), our bound inherits the result of Cortes et al. [8]. The cardinality of the hypothesis set $\mathcal{F} = \mathrm{conv}(\bigcup_{t=1}^T \mathcal{H}_t)$ is controlled by the mixture coefficients $\alpha_t$s in (1). $\sum_{t=1}^T \alpha_t |\mathcal{H}_t|$ in our bound implies that it is not detrimental to generalization if the corresponding mixture weight is relatively small, while some hypothesis sets used for learning could have large complexity. In other words, the coefficients $\alpha_t$s need to minimize the expected margin distribution loss $\mathbb{E}_S[\ell_{\mathrm{md}}(\sum_{l=1}^t \alpha_l \gamma_l(x))]$, which estimates the generalization error of the additive casForest model.

## 4   Optimization

The generalization analysis shows the importance of optimizing the margin ratio $\lambda$ and the mixture coefficients $\alpha_t$s. Since we formulate casForest as an additive model, we utilize the reweighting approach to minimize the expected margin distribution loss

$$\mathbb{E}_S \left[\ell_{\mathrm{md}} \left(\sum_{l=1}^t \alpha_l \gamma_l(x)\right)\right], \tag{26}$$

where the margin distribution loss function $\ell_{\mathrm{md}}$ is designed to utilize the first- and second-order statistics of margins, and $\gamma_l(x)$ denotes the margin in the $l$-th layer. The scalar $\alpha_t$ is determined by minimizing the expected loss for the $t$-layer model.

**The mdDF algorithm (Algorithm 2).** We denote a prediction score space by $C = \mathbb{R}^s$, where $s$ is the number of classes. When each sample passes through the forest model, we will get an average prediction vector in each layer: $h_t(\cdot) = \left[h_t^1(\cdot), h_t^2(\cdot) \ldots, h_t^s(\cdot)\right] \in C$. According to Crammer & Singer [10], we can define the margin of sample $\gamma_t(\cdot)$ for multi-class classification as: $\gamma_t(\cdot) := h_t^y(\cdot) - \max_{j \neq y} h_t^j(\cdot)$, i.e., the confidence of prediction.

The initial sample weights are $[1/m, 1/m, \ldots, 1/m]$, and we update the $i$-th weight by

$$\mathcal{D}_{t+1}(i) = \frac{\ell_{\mathrm{md}}\left(\sum_{l=1}^t \alpha_l \gamma_l(x_i)\right)}{\sum_{i=1}^m \ell_{\mathrm{md}}\left(\sum_{l=1}^t \alpha_l \gamma_l(x_i)\right)}, \tag{27}$$

where the margin distribution loss function $\ell_{\mathrm{md}}(\cdot)$ is defined by Zhang & Zhou [28] to optimize the first- and second-order statistics of margins as follows:

$$\ell_{\mathrm{md}}(z) = \begin{cases} \frac{(z-\gamma)^2}{\gamma^2} & z \leq \gamma, \\ \frac{\mu(z-\gamma)^2}{(1-\gamma)^2} & z > \gamma, \end{cases} \tag{28}$$

where hyper-parameter $\gamma$ is a parameter as the margin mean and $\mu$ is a parameter to trade off two different kinds of deviation (keeping the balance on both sides of the margin mean). Obviously, this margin distribution loss function will enforce the band that has a lower loss to contain the sample points as many as possible. In practice, we generally choose these two hyper-parameters from the finite sets $\gamma \in \{0.7, 0.75, 0.8, 0.85, 0.9, 0.95\}$ and $\mu \in \{0.01, 0.05, 0.1\}$. The algorithm utilizing the margin distribution optimization is summarized in Algorithm 2.

## 5 Experiments

**Datasets and configuration.** We choose eight classification benchmark datasets with different scales. The datasets vary in size: from 1484 up to 78823 instances, from 8 up to 784 features, and from 2 up to 26 classes. From the literature, these datasets come pre-divided into training and testing sets. Therefore in our experiments, we use them in their original format. PROTEIN, SENSIT, and SATIMAGE datasets are obtained from LIBSVM datasets [4]. Except for MNIST [18] dataset, others come from the UCI Machine Learning Repository [11]. Based on the attribute characteristics of the dataset, we classify the datasets into three categories: categorical, numerical, and mixed modeling tasks. We conjecture that some numerical modeling tasks such as image or audio recognition are very suitable for DNNs. Some operations, such as convolution, exactly fit well with numerical signal modeling. The deep forest model is not developed to replace DNNs for such tasks; instead, it offers an alternative when DNNs are not superior, e.g., deep forests are good at the categorical/symbolic or mixed modeling tasks especially [33].

In mdDF, we take two random forests and two completely-random forests in each layer, and each forest contains 100 trees, whose maximum depth of trees in random forests grows with the layer, i.e., $d_{max}^{(t)} \in \{2t+2, 4t+4, 8t+8, 16t+16\}$. To reduce the risk of overfitting, the representation learned by each forest is generated by $k$-fold cross-validation ($k = 5$ in our experiments). In Algorithm 1, each instance will be used as training data for $(k-1)$ times, and produce the final class vector as *augmented feature*s for the resulting in $(k-1)$ class vectors, that are averaged to the next layer.

We compare mdDF with the other four common used algorithms on different datasets: multilayer perceptron (MLP), random forest (RF) [3], XGBoost [5] and gcForest [32]. Here, we set the same number of forests as mdDF in each layer of gcForest. For random forests, we set $400 \times k$ trees; and for XGBoost, we also take $400 \times k$ trees. As for other hyper-parameters, we set them as the default values. For the multilayer perceptron (MLP) configurations, we use ReLU for the activation function, cross-entropy for the loss function, adadelta for optimization, no dropout for hidden layers according to the scale of training data. The network structure hyper-parameters, however, could not be fixed across tasks. Therefore, for MLP, we examine a variety of architectures on the validation set, and pick the one with the best performance, then train the whole network again on the training set and report the test accuracy. The examined architectures are listed as follows: (1) input-1024-512-output; (2) input-16-8-8-output; (3) input-70-50-output; (4) input-50-30-output; (5) input-30-20-output.

**Test accuracy on benchmark datasets.** Table 1 shows that mdDF achieves better accuracy than the other methods on several datasets. Compared with the MLP method, the deep forest models

Table 1: **Left:** Comparison results between mdDF and the other tree-based algorithms on test accuracy with different datasets. The best accuracy on each dataset is highlighted in bold type. • indicates the second best accuracy on each dataset. The average rank is listed at the bottom. **Right:** Comparison results between the standard mdDF structure and the other mdDF structures.

| Dataset | Attribute | MLP | RF | XGBoost | gcForest | mdDF | mdDF$_{SF}$ | mdDF$_{ST}$ | mdDF$_{NP}$ |
|---|---|---|---|---|---|---|---|---|---|
| ADULT | Categorical | 80.597 | 85.818 | 85.904 | 86.276 • | **86.560** | 86.200 | 85.710 | 85.650 |
| YEAST | Categorical | 59.641 | 61.886 | 59.161 | 63.004 • | **63.340** | 63.000 | 62.780 | 62.556 |
| LETTER | Categorical | 96.025 | 96.575 | 95.850 | 97.375 • | **97.500** | 96.475 | 97.300 | 96.975 |
| PROTEIN | Categorical | 68.660 | 68.071 | 71.214 • | 71.009 | **71.247** | 71.127 | 70.291 | 68.509 |
| HAR | Mixed | 94.231 • | 92.569 | 93.112 | 94.224 | **94.600** | 93.926 | 94.290 | 94.060 |
| SENSIT | Mixed | 78.957 | 80.133 | 81.874 | 82.334 • | **82.534** | 82.014 | 80.412 | 80.320 |
| SATIMAGE | Numerical | 91.125 | 91.200 | 90.450 | 91.700 • | **91.750** | 91.600 | 91.300 | 90.800 |
| MNIST | Numerical | 98.621 • | 96.831 | 97.730 | 98.252 | **98.734** | 98.254 | 98.101 | 98.240 |
| Avg. Rank | - | 3.650 | 4.000 | 3.750 | 2.375 | 1.000 | - | - | - |

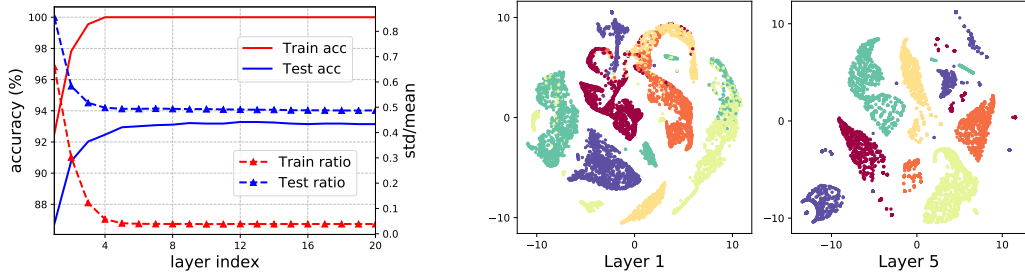

(a) The accuracy (solid line) and the margin ratio (dotted line) of the mdDF algorithm at different layers on the HAR dataset.

(b) The multi-layer feature visualization of the mdDF algorithm on HAR training set. The ratios of the intra-class variance to the inter-class variance $S_A/S_E$ are (3.88, 1.97).

Figure 2: The relation between the margin ratio and learning ability in the different layers.

almost outperform on these datasets and obtain the top 2 test accuracy on categorical or mixed modeling tasks. Obviously, gcForest and mdDF perform better than the shallow ones, and mdDF with reweighting and boosting representations outperforms gcForest across these datasets. The empirical results show that the deep models provide an improvement in performance with in-model transformation, compared to the shallow models that only have invariant features.

**Comparison with the other mdDF structures**   In Table 1, we compare our mdDF structure with the three other mdDF structures on different datasets: (1) mdDF using same forests (use 4 random forests) named mdDF$_{SF}$; (2) mdDF using stacking (only transmit the prediction vectors to next layer) named mdDF$_{ST}$; (3) mdDF without PRECONC (only transmit the input feature vector to next layer) named mdDF$_{NP}$. In this way, we explore the importance of internal structures of the mdDF. When we remove a concrete structure and control other variables, the performance of the mdDF algorithm will be worse. The empirical results demonstrate the effectiveness of these specific structures.

**Relation between the margin ratio and learning ability.**   Figure 2(a) plots the accuracy and margin ratio of mdDF on the HAR dataset. It demonstrates clearly that the performance is consistent with the margin ratio. When the margin ratio is smaller, i.e., the margin std/mean is smaller, the performance is better. Figure 2(b) plots the t-SNE visualization of mdDF on the HAR dataset. We also use the variance decomposition in the 2D space. The result shows that the intra-class compactness and inter-class separability are getting better as the layers becomes deeper. Such a correlation validates the theoretical result of our refined margin distribution analysis.

# 6   Conclusion

Recent studies propose a few tree-based deep models to learn the representations from a broad range of tasks and achieve good performance. By formulating casForest as an additive model, we partially explain the success of it from the perspective of the *margin theory*. The theoretical results inspire us to design a margin distribution reweighting approach that improves the generalization performance. Then, the empirical studies validate our theoretical results. We will explore how to understand the effectiveness of the PRECONC operation (which is crucial for feature enrichment) in future work.

**Acknowledgments**

This research was supported by the NSFC (61751306), National Key R&D Program of China (2018YFB1004300), and the Collaborative Innovation Center of Novel Software Technology and Industrialization. The authors would like to thank the anonymous reviewers for constructive suggestions, as well as Wei Gao, Lijun Zhang, Shengjun Huang, Xizhu Wu, Lu Wang, Peng Zhao, Ming Pang and Kangle Zhao for helpful discussions.

## Footnotes

[1]The hypothesis of the random forests block in the $t$-th layer contains that in the $(t-1)$-th layer without updating the augmented features, i.e., $\alpha_t = 0$. In other words, the in-model transformation [33] is crucial for the recursive formulation.

[2]In the binary classification, we can redefine the output of the strong classifier $F(x)$ as a variable in $[-1, 1]$, e.g. the difference between two prediction scores, where $\tilde{F}(x) = \text{sign}(F(x))$ is the predicted label. The previous bounds [8, 9, 17] are based on binary classification, therefore, our result is comparable with them.

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
