[Supplementary Material · Forest_sup.pdf]

# A    Notations

| | |
|---|---|
| $\mathcal{X}$ | The input domain |
| $\mathcal{Y}$ | The output domain |
| $\mathcal{D}$ | The underlying distribution over $\mathcal{X} \times \mathcal{Y}$ |
| $S$ | The training set of $m$ samples drawn according to $\mathcal{D}^m$ |
| $(x, y)$ | The sample drawn from the underlying distribution $\mathcal{D}$ |
| $h_t(x)$ | The $t$-layer CAScade Deep Forest (CASDF) |
| $\mathcal{H}_t$ | The hypotheses set of $t$-layer CASDF |
| $F(x)$ | The additive CASDF |
| $\tilde{\sigma}(\cdot)$ | The map from average prediction score vector to a label |
| $\tilde{F}(x)$ | The entire additive CASDF model |
| $\mathcal{F}$ | The convex hull of the union of $\mathcal{H}_1, \ldots, \mathcal{H}_T$ |
| $\mathcal{G}_{\mathcal{F},n}$ | The set of unweighted averages over $n$ elements from $\mathcal{F}$ |
| $\alpha_t$ | The scalar determined by optimizing $\ell_{\mathrm{md}}$ |
| $\ell_{\mathrm{md}}(x)$ | The convex margin distribution loss function |
| $[x, f_{t-1}(x)]$ | The concatenation operation of original features $x$ and augmented features $f_{t-1}(x)$ |
| $\phi_t([x, f_{t-1}(x)])$ | The function computed by forest block in the $t$-th layer |
| $f_t(x)$ | The augmented feature in the $t$-th layer |
| $\mathbb{E}_S[\cdot]$ | The empirical expectation over the training set |
| $\mathrm{Var}[\cdot]$ | The empirical variance over the training set |
| $\lambda$ | The margin ratio between the standard deviation and mean |
| $\mathcal{A}_{\mathrm{rfb}}$ | The random forest block algorithm that computes the function $g_t([x, f_{t-1}(x)])$ |
| $\mathcal{D}_t$ | The sample weight in the $t$-th layer |

# B    Optimal Margin Distribution Principle

Figure 3 shows that AdaBoost often does not overfit, that is, the test error often tends to decrease even after the training error reached zero, margin theory has been used to analyze this phenomenon which seemed to contradict with the Occam's Razor. We will introduce several algorithms based on margin theory, such as AdaBoost, Support Vector Machine (SVM) and Optimal Margin Distribution Machine (ODM).

Since Reyzin & Schapire [23] found that the **margin distribution** of AdaBoost is better than that of `arc-gv` [2] which is a boosting algorithm designed to maximize the minimum margin, Reyzin & Schapire [23] conjectured that margin distribution is more important to get a better generalization performance than the instance with the minimum margin. Gao & Zhou [16] prove that utilizing both the margin mean and margin variance can portray the relationship between margin and generalization performance for AdaBoost algorithm more precisely. We list the several loss functions of the algorithms based on margin theory to compare and plot them in Figure 4:

**Exponential loss function:**
$$\ell_{\exp}(x) = \exp\{-x\}. \tag{15}$$

**Hinge loss function:**
$$\ell_{\mathrm{hinge}}(x) = \max\{1 - x, 0\}. \tag{16}$$

Figure 3: An empirical result using AdaBoost with C4.5 decision trees as base learners. In this example, the training error goes to zero after about 5 rounds of boosting, yet the test error continues to decrease for larger values of $T$.

Figure 4: Examples of several convex upper bounds on the zero-one loss.

**Margin distribution loss function:**

$$\ell_{\mathrm{md}}(x) = \begin{cases} \frac{(x-1+\theta)^2}{(1-\theta)^2} & x \leq 1 - \theta, \\ 0 & 1 - \theta < x \leq 1 + \theta, \\ \frac{\mu(x-1-\theta)^2}{(1+\theta)^2} & x > 1 + \theta. \end{cases} \quad (17)$$

AdaBoost utilizes the exponential loss function to reweight the training samples in each round according to the margin distribution. This operation can make the model focus on dealing with the instance with a low confidence-rate (small margin). Mathematically, it proves that one version of the derivation of AdaBoost is achieved by minimizing the expected exponential loss function:

$$\ell_{\exp}(F|\mathcal{D}) = \mathbb{E}_{x\sim\mathcal{D}}\left[e^{-yF(x)}\right] \quad (19)$$

**Algorithm 3** AdaBoost Algorithm
***
**Input:** $(x_1, y_1), \ldots, (x_m, y_m)$ where $x_i \in \mathcal{X}, y_i \in \{-1, +1\}$.
**Output:** The additive model $H$.
  1: Initialize $D_1(i) = 1/m$ for $i = 1, \ldots, m$.
  2: **for** $t = 1, \ldots, T$ : **do**
  3:     Train weak learner using distribution $D_t$.
  4:     Get weak hypothesis $h_t : \mathcal{X} \to \{-1, +1\}$.
  5:     Aim: select ht to minimalize the weighted error:

$$\epsilon_t \doteq \Pr_{i \sim D_t} [h_t(x_i) \neq y_i]$$

.
  6:     Choose $\alpha_t = \frac{1}{2} \ln \left( \frac{1 - \epsilon_t}{\epsilon_t} \right)$.
  7:     Update, for $i = 1, \ldots, m$ :

$$
\begin{aligned}
D_{t+1}(i) &= \frac{D_t(i)}{Z_t} \times \left\{ \begin{array}{ll} e^{-\alpha_t} & \text{if } h_t(x_i) = y_i \\ e^{\alpha_t} & \text{if } h_t(x_i) \neq y_i \end{array} \right. \\
&= \frac{D_t(i) \exp(-\alpha_t y_i h_t(x_i))}{Z_t},
\end{aligned}
\tag{18}
$$

      where $Z_t$ is a normalization factor (chosen so that $D_{t+1}$ will be a distribution).
  8: **end for**
  9: Output the final hypothesis:

$$F(x) = \text{sign}\left( \sum_{t=1}^{T} \alpha_t h_t(x) \right).$$
***

using the additive weighted combination of weak learners as:

$$F(x) = \sum_{t=1}^{T} \alpha_t h_t(x) \tag{20}$$

The exponential loss is used here since it gives an elegant update formula, and it is consistent with the goal of minimizing classification error. Schapire & Freund [24] give theoretical evidence that AdaBoost is especially suited to the task of maximizing the number of training examples with a large margin. Informally, this is because, at every round, AdaBoost puts the most weight on the examples with the smallest margins.

**Theorem 2** (Schapire & Freund [24])**.** *Given the notation of AdaBoost Algorithm 3, let $\gamma_t = \frac{1}{2} - \epsilon_t$. Then the fraction of training examples with margin at most $r$ is at most*

$$\prod_{t=1}^{T} \sqrt{(1 + 2\gamma_t)^{1+r} (1 - 2\gamma_t)^{1-r}}. \tag{21}$$

When $\sqrt{(1 - 2\gamma)^{1-r}(1 + 2\gamma)^{1+r}} < 1$, this bound implies that the fraction of training examples with $yf(x) \leq r$ decreases to zero exponentially fast with $T$, and must actually be equal to zero at some point since this fraction must always be $\mathcal{O}(1/m)$. In a word, optimizing the exponential loss function for classification tasks can get a classifier with good margin distribution.

Boosting is not the only classification method that seems to operate on the principle of margin maximization. In particular, Support Vector Machines (SVMs), which are based explicitly on this principle, are currently very popular due to their effectiveness for general machine-learning tasks. SVMs can be formulated as optimizing a hinge loss with $\ell_2$ regularization. However, SVMs focus on maximizing the minimum margin which is similar to the `arc-gv` algorithm. Although such methods may succeed at increasing the minimum margin among all training examples, this increase may come at the expense of the vast majority of the other training examples, so that although the minimum margin increases, the bulk of the margin distribution decreases.

Figure 5: A simple illustration of linear separators optimizing the minimum margin, margin mean and margin distribution, respectively. $h_{\min}$ represents the classifier learned by maximizing the minimum margin. $h_{\mathrm{mean}}$ represents that learned by maximizing the margin mean. $h_{\mathrm{dist}}$ represents that learned by optimizing the margin distribution through maximizing the margin variance and minimizing the margin variance simultaneously.

Compared with maximizing the minimum margin, the optimal margin distribution principle [16, 28] conjecture that maximizing the margin mean and minimizing the margin variance is the key to achieving a better generalization performance. Figure 5 shows that optimizing the margin distribution with first- and second-order statistics can utilize more information on training data, e.g. the covariance among the different features. Inspired by this idea, Zhang & Zhou [28] propose the optimal margin distribution machine (ODM) which can be formulated as:

$$
\begin{aligned}
\min_{\boldsymbol{w},\xi_i,\epsilon_i} \quad & \Omega(\boldsymbol{w}) + \frac{\lambda}{m}\sum_{i=1}^{m}\frac{\xi_i^2 + \mu\epsilon_i^2}{(1-\theta)^2} \\
\text{s.t.} \quad & \gamma_h\left(\boldsymbol{x}_i,y_i\right) \geq 1 - \theta - \xi_i \\
& \gamma_h\left(\boldsymbol{x}_i,y_i\right) \leq 1 + \theta + \epsilon_i, \forall i
\end{aligned}
\tag{22}
$$

where $\theta + \xi_i$ and $\theta + \epsilon_i$ are the deviation of the margin $\gamma_h(x_i,y_i)$ to the margin mean, $\mu \in (0,1]$ is a parameter to trade off two different kinds of deviation (larger or less than margin mean). $\theta \in [0,1)$ is a parameter of the **zero-loss band**, which can control the number of support vectors, i.e., the sparsity of the solution, and $(1-\theta)^2$ in the denominator is to scale the second term to be a surrogate loss for 0-1 loss. Similar to support vector machines (SVMs), we can give ODM an intuitive illustration in Figure 6. Similar to formulating support vector machines as a combination of the hinge loss and the regularization term, we can use margin distribution loss function $\ell_{\mathrm{md}}$ defined in (17) and a regularization term to represent the ODM. The simplified version margin distribution loss function (14) is similar to that of the ODM. Our forest representation learning approach requires as many samples as possible to train the model and generate the augmented features. Therefore, we remove the parameter $\theta$ which can control the number of support vectors. Our loss function is to optimize the margin distribution to minimize the margin ratio $\lambda$, referring to Remark 2 in Section 3.

## C   Complete Proofs for Section 3

### C.1   Preliminaries

For simplicity, we consider the binary classification task. We define the strong classifier as $F(x) = \sum_{t=1}^{T}\alpha_t h_t$, i.e., CASDF is formulated as an additive model. Now we define the margin for sample $(x,y)$ as $yF(x) \in [-1,1]$, which implies the confidence of prediction. We assume that the hypotheses set $\mathcal{H}$ of base classifiers $\{h_1, h_2, \ldots, h_T\}$ can be decomposed as the union of $T$ families $\mathcal{H}_1, \mathcal{H}_2, \ldots, \mathcal{H}_T$ ordered by increasing complexity, where $\forall t \geq 1, \mathcal{H}_t \subset \mathcal{H}_{t+1}$ and $h_t \in \mathcal{H}_t$.

Figure 6: A simple illustration of optimal margin distribution machine (ODM). We assume that the margin mean is preset to a constant 1, so that $\theta$ is somewhat a parameter which implies the margin variance. Since the sample points away form margin mean, i.e., $\xi_i > 0 \vee \epsilon_i > 0$, will be imposed a square-type penalty, $[1 - \theta, 1 + \theta]$ is the zero-loss band to contain as much training data as possible. At last, when we minimize the margin distribution loss with a regularization term $\Omega(\boldsymbol{w})$, we maximize the normalized margin mean $\frac{1}{\Omega(\boldsymbol{w})}$ with a margin variance controlled by parameter $\theta$.

Remarkably, the complexity term admits an explicit dependency in terms of the mixture coefficients defining the ensembles. Thus, the ensemble family we consider is $\mathcal{F} = \mathrm{conv}\left(\bigcup_{t=1}^{T} \mathcal{H}_t\right)$, which is the family of functions $F(x)$ of the form $F(x) = \sum_{t=1}^{T} \alpha_t h_t(x)$, where $\boldsymbol{\alpha} = (\alpha_1, \ldots, \alpha_T)$ is in the simplex $\Delta$.

For a fixed $\mathbf{g} = (1, \ldots, g_T)$, any $\boldsymbol{\alpha} \in \Delta$ defines a distribution over $\{g_1, \ldots, g_T\}$. Sampling from $\{g_1, \ldots, g_T\}$ according to $\boldsymbol{\alpha}$ and averaging leads to functions $G = \frac{1}{n} \sum_{i=1}^{T} n_t g_t$ for some $\mathbf{n} = (n_1, \ldots, n_T)$, with $\sum_{t=1}^{T} n_t = n$, and $g_t \in \mathcal{H}_t$. For any $\mathbf{N} = (N_1, \ldots, N_T)$ with $|\mathbf{N}| = n$, we consider the family of functions

$$\mathcal{G}_{\mathcal{F},\mathbf{N}} = \left\{ \frac{1}{n} \sum_{k=1}^{T} \sum_{j=1}^{N_k} g_{k,j} \, \middle| \, \forall (k,j) \in [T] \times [N_k], g_{k,j} \in \mathcal{H}_k \right\}, \tag{23}$$

and the union of all such families $\mathcal{G}_{\mathcal{F},n} = \bigcup_{|\mathbf{N}=n|} \mathcal{G}_{\mathcal{F},\mathbf{N}}$. For a fixed $\mathbf{N}$, the size of $\mathcal{G}_{\mathcal{F},\mathbf{N}}$ can be bounded as follows:

$$\ln |\mathcal{G}_{\mathcal{F},\mathbf{N}}| \leq \ln \left( \prod_{t=1}^{T} |\mathcal{H}_t|^{N_t} \right) = \sum_{t=1}^{T} (N_t \ln |\mathcal{H}_t|) = n \sum_{t=1}^{T} (\alpha_t \ln |\mathcal{H}_t|) \leq n \ln \sum_{t=1}^{T} \alpha_t |\mathcal{H}_t| \tag{24}$$

**Technical lemmas:**

**Lemma 2** (Chernoff bound [6]). *Let $X, X_1, \ldots, X_m$ be $m+1$ i.i.d. random variables with $X \in [0,1]$. Then, for any $\epsilon > 0$, we have*

$$\Pr[\frac{1}{m} \sum_{i=1}^{m} X_i \geq \mathbb{E}[X] + \epsilon] \leq \exp\left(-\frac{m\epsilon^2}{2}\right), \tag{25}$$

$$\Pr[\frac{1}{m} \sum_{i=1}^{m} X_i \leq \mathbb{E}[X] - \epsilon] \leq \exp\left(-\frac{m\epsilon^2}{2}\right). \tag{26}$$

**Lemma 3** (Gao & Zhou [16]). *For independent random variables $X_1, X_2, \ldots, X_m (m \geq 5)$ with values in $[0,1]$, and for $\delta \in (0,1)$, with probability at least $1 - \delta$ we have*

$$\frac{1}{m} \sum_{i=1}^{m} \mathbb{E}[X_i] - \frac{1}{m} \sum_{i=1}^{m} X_i \leq \sqrt{\frac{2\hat{V}_m \ln(2/\delta)}{m}} + \frac{7 \ln(2/\delta)}{3m}, \tag{27}$$

$$\frac{1}{m}\sum_{i=1}^{m}\mathbb{E}[X_i] - \frac{1}{m}\sum_{i=1}^{m}X_i \geq -\sqrt{\frac{2\hat{V}_m\ln(2/\delta)}{m}} - \frac{7\ln(2/\delta)}{3m}. \tag{28}$$

*where* $\hat{V}_m = \sum_{i\neq j}(X_i - X_j)^2/2m(m-1)$

## C.2   Proofs of Lemma 1 and Theorem 1

*Proof of Lemma 1.* For $\lambda > 0$, according to the Markov's inequality, we have

$$\Pr_{S,\mathcal{G}_{\mathcal{F},n}}[yG(x) - yF(x) \geq \epsilon] = \Pr_{S,\mathcal{G}_{\mathcal{F},n}}[(yG(x) - yF(x))n\lambda/2 \geq n\lambda\epsilon/2] \tag{29}$$

$$\leq \exp\left(-\frac{\lambda n\epsilon}{2}\right)\mathbb{E}_{S,G_j\in\mathcal{G}_{\mathcal{F},n}}\left[\exp\left(\frac{\lambda}{2}\sum_{j=1}^{n}(yG_j(x) - yF(x))\right)\right] \tag{30}$$

$$= \exp\left(-\frac{\lambda n\epsilon}{2}\right)\prod_{j=1}^{n}\mathbb{E}_{S,G_j\in\mathcal{G}_{\mathcal{F},n}}\left[\exp\left(\frac{\lambda}{2}(yG_j(x) - yF(x))\right)\right] \tag{31}$$

where the last inequality holds from the independent of $G_i$. Notice that $|yG_j(x) - yF(x)| \leq 2$ (the margin is bounded: $yF(x) \in [-1, 1]$), using Taylor's expansion, we get

$$\mathbb{E}_{S,G_j\in\mathcal{G}_{\mathcal{F},n}}\left[\exp\left(\frac{\lambda}{2}(yG_j(x) - yF(x))\right)\right] \leq 1 + \mathbb{E}_{S,G_j\in\mathcal{G}_{\mathcal{F},n}}[(yG_j(x) - yF(x))^2]\frac{e^\lambda - 1 - \lambda}{4} \tag{32}$$

$$\leq 1 + \mathbb{E}_S[1 - (yF(x))^2]\frac{e^\lambda - 1 - \lambda}{4} \tag{33}$$

$$\leq \exp(1 - \mathbb{E}_S^2[yF(x)])\frac{e^\lambda - 1 - \lambda}{4} \tag{34}$$

where the last inequality holds from Jensen's inequality and $1 + x \leq e^x$. Therefore, we have

$$\Pr_{S,\mathcal{G}_{\mathcal{F},n}}[yG(x) - yF(x) \geq \epsilon] \leq \exp\left(\frac{n(e^\lambda - 1 - \lambda)(1 - \mathbb{E}_S[yF(x)])}{4} - \frac{\lambda n\epsilon}{2}\right) \tag{35}$$

If $0 < \lambda < 3$, then we could use Taylor's expansion again to have

$$e^\lambda - \lambda - 1 = \sum_{i=2}^{\infty}\frac{\lambda^i}{i!} \leq \frac{\lambda^2}{2}\sum_{m=0}^{\infty}\frac{\lambda^m}{3^m} = \frac{\lambda^2}{2(1 - \lambda/3)}. \tag{36}$$

Now by picking $\lambda = \frac{\epsilon}{1/2 - \mathbb{E}_S^2[yF(x)]/2 + \epsilon/3}$, we have

$$-\frac{\lambda\epsilon}{2} + \frac{\lambda^2(1 - \mathbb{E}_S^2[yF(x)])}{8(1 - \lambda/3)} \leq \frac{-\epsilon^2}{2 - 2\mathbb{E}_S^2[yF(x)] + 4\epsilon/3} \tag{37}$$

By Combining (35) and (37) together, we complete the proof. $\qquad\square$

*Proof of Theorem 1.* For $F = \sum_{t=1}^{T}\alpha_t h_t \in \mathcal{F}$ and $G \in \mathcal{G}_{\mathcal{F},n}$, we have $\mathbb{E}_{G\in\mathcal{G}_{\mathcal{F},n}}[G] = F$. For $\beta > 0$, the Chernoff's bound in Lemma 2 gives

$$\Pr_D[yF(x) < 0] = \Pr_{D,\mathcal{G}_{\mathcal{F},n}}[yF(x) < 0, yG(x) \geq \beta] + \Pr_{D,\mathcal{G}_{\mathcal{F},n}}[yF(x) < 0, yG(x) < \beta] \tag{38}$$

$$\leq \exp(-n\beta^2/2) + \Pr_{D,\mathcal{G}_{\mathcal{F},n}}[yG(x) < \beta]. \tag{39}$$

Recall that $|\mathcal{G}_{\mathcal{F},N}| \leq \prod_{t=1}^{T}|\mathcal{H}_t|^{N_t}$ for a fixed $N$. Therefore, for any $\delta_n > 0$, combining the union bound with Lemma 3 guarantees that with probability at least $1 - \delta_n$ over sample $S$, for any $G \in \mathcal{G}_{\mathcal{F},N}$ and $\beta > 0$

$$\Pr_D[yG(x) < \beta] \leq \Pr_S[yG(x) < \beta] + \sqrt{\frac{2}{m}\hat{V}_m\ln\left(\frac{2}{\delta}\prod_{t=1}^{T}|\mathcal{H}_t|^{N_t}\right)} + \frac{7}{3m}\ln\left(\frac{2}{\delta}\prod_{t=1}^{T}|\mathcal{H}_t|^{N_t}\right) \tag{40}$$

$$= \Pr_S[yG(x) < \beta] + \sqrt{\frac{2}{m}\hat{V}_m \sum_{i=1}^{T} N_t \ln\left(\frac{2|\mathcal{H}_t|}{\delta}\right)} + \frac{7}{3m}\sum_{i=1}^{T} N_t \ln\left(\frac{2|\mathcal{H}_t|}{\delta}\right) \tag{41}$$

$$\leq \Pr_S[yG(x) < \beta] + \sqrt{\frac{2n}{m}\hat{V}_m \sum_{i=1}^{T} \alpha_t \ln\left(\frac{2|\mathcal{H}_t|}{\delta}\right)} + \frac{7n}{3m}\sum_{i=1}^{T} \alpha_t \ln\left(\frac{2|\mathcal{H}_t|}{\delta}\right) \tag{42}$$

$$\leq \Pr_S[yG(x) < \beta] + \sqrt{\frac{2n}{m}\hat{V}_m \ln\left(\frac{2\sum_{i=1}^{T}\alpha_t|\mathcal{H}_t|}{\delta}\right)} + \frac{7n}{3m} \ln\left(\frac{2\sum_{i=1}^{T}\alpha_t|\mathcal{H}_t|}{\delta}\right) \tag{43}$$

$$\tag{44}$$

where

$$\hat{V}_m = \sum_{i \neq j} \frac{(\mathbb{I}[y_iG(x_i) < \beta] - \mathbb{I}[y_jG(x_j) < \beta])^2}{2m(m-1)}, \tag{45}$$

The inequality (42) is a large probability bound when $n$ is large enough and inequality (43) is according to the Jensen's Inequality. Since there are $T$ at most $T^n$ possible $T$-tuples $N$ with $|N| = n$, by the union bound, for any $\delta > 0$, with probability at least $1 - \delta$, for all $G \in \mathcal{G}_{\mathcal{F},n}$ and $\beta > 0$:

$$\Pr_D[yG(x) < \beta] \leq \Pr_S[yG(x) < \beta] + \sqrt{\frac{2n}{m}\hat{V}_m \ln\left(\frac{2\sum_{i=1}^{T}\alpha_t|\mathcal{H}_t|}{\delta/T^n}\right)} + \frac{7n}{3m} \ln\left(\frac{2\sum_{i=1}^{T}\alpha_t|\mathcal{H}_t|}{\delta/T^n}\right) \tag{46}$$

Meantime, we can rewrite $\hat{V}_m$

$$\hat{V}_m = \sum_{i \neq j} \frac{(\mathbb{I}[y_iG(x_i) < \beta] - \mathbb{I}[y_jG(x_j) < \beta])^2}{2m(m-1)} \tag{47}$$

$$= \frac{2m^2 \Pr_S[yG(x) < \beta] \Pr_S[yG(x) \geq \beta]}{2m(m-1)} \tag{48}$$

$$= \frac{m}{m-1}\hat{V}_m^* \tag{49}$$

For any $\theta_1, \theta_2 > 0$, we utilize Chernoff's bound in Lemma 3 to get:

$$\hat{V}_m^* = \Pr_S[yG(x) < \beta] \Pr_S[yG(x) \geq \beta] \tag{50}$$

$$\leq 3\exp(-n\theta_1^2/2) + \Pr_S[yF(x) < \beta + \theta_1] \Pr_S[yF(x) \geq \beta - \theta_1] \tag{51}$$

$$\leq 3\exp(-n\theta_1^2/2) + \Pr_S[yF(x) < \beta + \theta_1 \,|\, \mathbb{E}_S[yF(x)] \geq \beta + \theta_1 + \theta_2] \tag{52}$$

$$\quad \cdot \Pr_S[yF(x) \geq \beta - \theta_1 | \mathbb{E}_S[yF(x)] \geq \beta + \theta_1 + \theta_2]$$

$$\leq 3\exp(-n\theta_1^2/2) + \frac{\text{Var}[yF(x)]}{\theta_2^2} \qquad \text{(According to Chebyshev's Inequality)}$$

$$\leq 3\exp(-n\theta_1^2/2) + \frac{\text{Var}[yF(x)]}{(\mathbb{E}_S[yF(x)] - \beta + \theta_1)^2} \tag{53}$$

$$\simeq 3\exp(-n\theta_1^2/2) + \frac{\text{Var}[yF(x)]}{\mathbb{E}_S^2[yF(x)]} \tag{54}$$

where $\text{Var}[yF(x)] = \mathbb{E}_S[(yF(x))^2] - \mathbb{E}_S^2[yF(x)]$ is the variance of the margins.

From Lemma 1, we obtain that

$$\Pr_S[yG(x) < \beta] \leq \Pr_S[yF(x) < \beta + \theta_1] + \exp\left(\frac{-n\theta_1^2}{2 - 2\mathbb{E}_S^2[yF(x)] + 4\theta_1/3}\right) \tag{55}$$

Let $\theta_1 = r/6$, $\beta = 5r/6$ and $n = \ln m/r^2$, then we combine (39), (40), (54) and (55), the proof is completed. $\qquad\square$

## D   Extended Experiments

### D.1   Comparison with the other mdDF structures

Here we compare our mdDF with three other mdDF structures on different datasets: (1) mdDF using same forests (use 4 random forests), i.e., mdDF$_{\text{SF}}$; (2) mdDF using stacking (only transmit the prediction vectors to next layer), i.e., mdDF$_{\text{ST}}$; (3) mdDF without PRECONC (only transmit the input feature vector to next layer), i.e., mdDF$_{\text{NP}}$. In this way, we explore the importance of internal structures (different types of forest and PRECONC operation) of the mdDF.

The experimental results are in line with our expectations. When we remove a concrete structure while controlling other variables, the generalization performance of the mdDF approach will be worse. This empirical result demonstrates the effectiveness of these specific model structures.

### D.2   Margin ratio and feature visualization

**Relation between the margin ratio and learning ability.** We conduct experiments on the HAR, PROTEIN, MNIST and ADULT data sets with mdDF and record the changes in accuracy when the layer increases in Figure 7. Similar to Schapire et al. [25], we plot the changes in accuracy. It is clear that mdDF achieves 100% accuracy on training sets in less than 3 layers and stays the same ever since. However, the test accuracy keeps increasing when the layer increasing more than 3 layers. As we show in Theorem. 1, the margin distribution is crucial to explain why the algorithm seems resistant to the overfitting problem. Especially, we can evaluate the margin distribution by calculating the ratio of margin variance to square of margin mean. We show that the margin ratio $\lambda$ and distribution varies with the layers, i.e., the margin mean becomes larger and the margin variance become smaller in Figure 7.

Since the performance of mdDF model is excellent, we hope to see that the distributions of data in the learned feature space (in different layers) are consistent with the generalization ability. In this experiment, we use the t-SNE method to visualize the data distribution in different layers for training samples and test samples. Figure 8 plots the 2-dimension embedding image on several high-dimensional datasets. The t-SNE [21] is a tool to visualize high-dimensional data. It converts similarities between data points to joint probabilities and tries to minimize the Kullback-Leibler divergence between the joint probabilities of the low-dimensional embedding and the high-dimensional data.

All these experiments and visualizations confirm the effectiveness of mdDF in terms of performance and representation learning ability. Furthermore, the empirical results show the correlation between margin ratio and generalization performance. We can find that the visualization of mdDF is getting better as the layer becomes deeper, the intra-class compactness and inter-class separability of learned feature space are getting better. To quantify the degree of compactness of the distribution, we perform a variance decomposition on the data in the embedding space. Such a correlation validates the theoretical result of our refined margin distribution analysis.

Table 2: Comparison results between mdDF and the other mdDF structures on test accuracy with different datasets.

| Dataset | mdDF$_{\text{SF}}$ | mdDF$_{\text{ST}}$ | mdDF$_{\text{NP}}$ | mdDF |
|---|---|---|---|---|
| ADULT | 86.200 | 85.710 | 85.650 | **86.560** |
| YEAST | 63.000 | 62.780 | 62.556 | **63.340** |
| LETTER | 96.475 | 97.300 | 96.975 | **97.500** |
| PROTEIN | 71.127 | 70.291 | 68.509 | **71.247** |
| HAR | 93.926 | 94.290 | 94.060 | **94.600** |
| SENSIT | 82.014 | 80.412 | 80.320 | **82.534** |
| SATIMAGE | 91.600 | 91.300 | 90.800 | **91.750** |
| MNIST | 98.254 | 98.101 | 98.240 | **98.734** |

(a)

(b)

(c)

(d)

(e)

(f)

(g)

(h)

Figure 7: The training & test accuracy (solid) and margin ratio (dot-dashed) of mdDF model in different layers on the HAR (a), PROTEIN (c), MNIST (e) and ADULT (g) data sets . Margin rate in different layers on the UCI HAR data set. Margin distribution of mdDF model on the HAR (b), PROTEIN (d), MNIST (f) and ADULT (h) data sets .

Figure 8: Multi-layer feature visualization of mdDF on HAR training set (a), test set (b), PROTEIN training set (c), test set (d), MNIST training set (e) and test set (f). We do the variance decomposition in this 2D space, and the ratio of the intra-class variance to the inter-class variance $S_A/S_E$ can be obtained as follows: (a) [3.88, 1.97, 0.72, 0.65], (b) [1.69, 0.88, 0.75, 0.51], (c) [15.12, 10.02, 7.5, 1.3], (d) [16.01, 14.85, 12.33, 9.86], (e) [0.43, 0.38, 0.15, 0.03], (f) [0.45, 0.43, 0.21, 0.11], i.e., the intra-class compactness and inter-class separability is getting better as the layer becomes deeper. Extensive margin distribution results are shown as a curve in Figure 7 correspondingly.