[Reviews · NeurIPS 2019]

Reviewer 1



Originality: To the best of my knowledge, the work presented in this paper is quite novel. Clarity: The paper is well written and well structured. Quality: The paper advances the understanding of CasDF with strong well proved theoretical results. Moreover, the derived algorithm makes sense, and the experimental analysis is strong enough. Significance: The theoretical result of the paper is in my opinion important and useful to the field. The experiments show improvement over the baselines in all setting. One missing detail in the results is to report average performance with standard deviation to show the significance of this improvement, especially that some results are close to each other.

Reviewer 2



[response to authors] I’ve read all reviewers’ comments and the authors’ response letter. I think this work has theoretical novelty in understanding CASDF and has experimental support. It will be valuable to the field for future studies. Comments are addressed. Overall, I will keep my score. -------------------------- This paper provides a new perspective to understand and explain the cascade deep forest, and proposes a margin distribution reweighting approach to minimize the gap between the generalization error and empirical margin loss, which produces much better model performance. The method is supported by mathematical theories and empirical experiments. The theories and proofs are clear and solid. Experiment settings and results are clear.

Reviewer 3



Originality: The primary contribution is a tighter bound. From skimming, this comes from using a bernstein style bound so when the variance is small, there can be a tighter rate. Quality: The paper is reasonably well written and the claims appear correct. I did not check the proofs. Clarity: The paper is reasonably well-organized. It is still difficult to follow because it requires the reader to be familiar with Zhou and Feng though it is concisely summarized. Significance: Unclear. The primary contribution seems to be an improved boosting style algorithm that removes a sqrt. The experiments are completely unconvincing.

[Author Response · NeurIPS 2019]

We want to thank reviewers for insightful comments, and we will improve the paper accordingly. In the following, we focus on technical questions.

## Reviewer 1

**Q1:** "One missing detail in the results is to report average performance with standard deviation to show the significance of this improvement, especially that some results are close to each other."

**A1:** We will add the standard deviation and error bars.

Thanks for your appreciation on this work.

## Reviewer 3

**Q1:** "There are a lot of notations in this paper, it will be easier to read and follow if there is a single table to explain the meaning of those notations"

**A1:** We will add a table to summarize and explain all notations.

**Q2:** "Will you plan to publish the code of experiments?"

**A2:** We will publish the code after the acceptance of this work.

Thanks for your appreciation on this work.

## Reviewer 4

**Q1:** "The experiments . . . It seems to be mostly on UCI datasets"

**A1:** MNIST is not UCI data. We use these datasets simply because they have been popularly used as baselines for forest approaches [3,5,31]. We will add some non-UCI datasets, e.g.,

| Dataset | #Instance | MLP | RF | XGBoost | gcForest | MDDF |
|---------|-----------|-----|-----|---------|----------|------|
| *ijcnn1* | 141,691 | 98.391±0.056 | 98.567±0.028 | 99.133±0.024 | 99.404±0.026● | **99.427±0.041** |
| *webspam* | 350,000 | 98.997±0.039 | 98.795±0.021 | 99.105±0.017 | 99.274±0.033● | **99.289±0.038** |
| . . . . . . | | | | | | |

[Meta-Review · NeurIPS 2019]

This paper formulates a forest representation learning approach (CASDF) as an additive model which boosts the augmented features and improves the upper bound on the generalization gap (by removing square root). The analysis results in a margin distribution reweighting scheme for deep forests. The paper is well-written, and is well-organized. Also the majority of reviewers find the contributions in the paper significant. However, R3 finds the experiments in the paper very limited. Nevertheless, given that the main emphasis and contribution of the paper is in theoretical analysis, I am fine with limited experimentation. Considering overall rating of this work, I recommend accept.